# Effect of Manual Massage, Foam Rolling, and Strength Training on Hemodynamic and Autonomic Responses in Adults: A Scoping Review

**DOI:** 10.3390/healthcare13121371

**Published:** 2025-06-07

**Authors:** Estêvão Rios Monteiro, Lavínia Martins Aguilera, Maria Ruá-Alonso, Gleisson da Silva Araújo, Victor Gonçalves Corrêa Neto, Cláudio Melibeu Bentes, José Vilaça-Alves, Victor Machado Reis, Arthur de Sá Ferreira, Paulo H. Marchetti, Jefferson da Silva Novaes

**Affiliations:** 1Postgraduate Program in Physical Education, Universidade Federal do Rio De Janeiro (UFRJ), Rio de Janeiro 21941-599, Brazil; profgleisson@ufrj.br (G.d.S.A.); jeffsnovaes@gmail.com (J.d.S.N.); 2Postgraduate Program in Rehabilitation Sciences, Centro Universitário Augusto Motta (UNISUAM), Rio de Janeiro 21032-060, Brazil; laviniaaguilera@souunisuam.com.br (L.M.A.); asferreira@unisuam.edu.br (A.d.S.F.); 3Postgraduate Program in Biopsychosocial Health, Centro Universitário Augusto Motta (UNISUAM), Rio de Janeiro 21041-020, Brazil; 4Performance and Health Group, Department of Physical Education and Sport, Faculty of Sports Sciences and Physical Education, University of A Coruna, 15179 A Coruña, Spain; maria.rua@udc.es; 5Postgraduate Program in Human Movement Sciences, School of Physical Education, Physiotherapy and Dance, Federal University of Rio Grande do Sul (UFRGS), Porto Alegre 90690-200, Brazil; 6School of Physical Education and Physiotherapy, Federal University of Pelotas (UFPel), Pelotas 96055-630, Brazil; 7Undergraduate Program in Physical Education, Universidade Federal do Rio de Janeiro (UFRJ), Rio de Janeiro 23890-000, Brazil; profvictorneto@gamaesouza.com; 8Laboratory of Physiology and Human Performance, Department of Physical Education and Sports, Institute of Education, Federal Rural University of Rio de Janeiro, Seropédica 23890-000, Brazil; claudiomelibeu@ufrrj.br; 9Physical Activity Sciences Graduate Program, Salgado de Oliveira University (UNIVERSO), Niterói 24030-060, Brazil; 10Department of Sports Sciences, Exercise and Health, Universidade de Trás-os-Montes e Alto Douro, 5000-801 Vila Real, Portugal; josevilaca@utad.pt (J.V.-A.); vmreis@utad.pt (V.M.R.); 11Research Center in Sports Sciences, Health Sciences and Human Development (CIDESD), 5001-801 Vila Real, Portugal; 12Resistance Training Laboratory, California State University, Northridge, CA 91330, USA; paulo.marchetti@csun.edu; 13Strength Training Laboratory (LABFOR), Federal University of Juiz de Fora, São Pedro 36036-900, Brazil

**Keywords:** blood physiological, post-exercise hypotension, myofascial release therapy, manual therapy

## Abstract

**Objectives:** This review explores the current evidence on how different massage modalities, either manual (MM) or using foam rolling (FR), with or without strength training, influence cardiovascular and autonomic function in healthy individuals. **Methods**: A search was performed in CINAHL, Cochrane Library, PubMed^®^, and SciELO databases on 14 April 2025. **Results**: Among the 5125 studies retrieved in the database search, 7 were selected for the present review. The included studies pointed to an improvement in hemodynamic and autonomic responses, characterized by reduced arterial stiffness and blood pressure and an increase in nitric oxide concentration and blood flow. These findings suggest that physical exercise prescribers should consider the hemodynamic and autonomic effects promoted by massage (MM or FR). **Conclusions**: A change in arterial compliance, followed by a hypotensive effect on systolic blood pressure, reinforces the role of physical activity as a non-pharmacological agent and highlights the need for inclusion in the different groups that need adjuvant help for blood pressure control.

## 1. Introduction

Elevated blood pressure (BP) over time is a well-established risk factor for cardiovascular events [1,2], making it a significant global public health concern [3,4]. As a result, various non-pharmacological approaches aimed at acutely reducing BP have been widely explored. The American College of Sports Medicine [5] highlights regular physical activity as a fundamental intervention for enhancing overall health and achieving both immediate and long-term BP reductions [6], including hypertension prevention in individuals with optimal BP levels [7]. Among the diverse strategies investigated to induce post-exercise hypotension (PEH) [8], aerobic exercise remains the most extensively studied and recommended modality [9]. However, growing evidence has emerged regarding the role of strength training (ST), demonstrating that even a single ST session can lead to a reduction in resting BP following exercise [10].

Similar PEH effects are observed when using foam rolling (FR) [11,12] or manual massage (MM) [13,14]. In this way, Liao et al. [13] conducted a systematic review with meta-analysis and observed that the classic MM may promote PEH in systolic BP (−7.39 mmHg; effect size = −0.728) and diastolic BP (−5.04 mmHg; effect size = −0.334). The similarity of these findings suggests equivalent effects between FR and MM, despite the technical differences in their application. Among these differences, it is noteworthy that the superficial contact with the participant’s skin and the application of sliding pressure on the tissue appear to stimulate muscular and fascial mechanoreceptors, which exert inhibitory effects, ultimately leading to a reduction in muscle tone [14]. This reduced muscle tone can promote a shift from sympathetic to parasympathetic dominance, facilitating the processes for PEH to occur. Thus, FR and MM could be a useful tool to acutely reduce BP values.

Understanding the effects of these techniques (FR and MM), whether applied in isolation or combined with ST, helps bridge gaps in exercise prescription and enhances the plausibility of their integration into treatment routines (through movement) targeting autonomic and hemodynamic responses. However, there is limited evidence in the literature regarding the blood pressure response to different combinations of FR and ST. The initial BP response to exercise is well documented [15]. A transient acute increase in BP is observed during physical exertion [16,17], resulting from the complex interplay of regulatory mechanisms that support the heightened energy demand [18,19,20] and counteract the substantial acute rise in total peripheral resistance [17]. Following this transient elevation, BP typically returns to baseline or even falls below pre-exercise levels [21], largely influenced by changes in vascular resistance and other physiological adjustments. However, while these immediate and short-term BP reductions are well recognized, the later phases of BP recovery (>30 min post-exercise) have received less attention in the literature, particularly in female populations. Nonetheless, evidence suggests that a delayed BP reduction may occur.

Since ST and FR elicit distinct initial stimuli within the nervous system related to acute BP regulation, their combination may produce additive effects on BP responses. If this synergistic interaction is confirmed in normotensive individuals, it could highlight an additional advantage of incorporating FR into an ST regimen, particularly when the prescribed ST volume necessary to elicit a hypotensive effect cannot be fully achieved. Thus, understanding the isolated and combined effects of these interventions on autonomic and hemodynamic responses may provide exercise professionals (e.g., Exercise Physiologists, Biomechanists, and Physical Therapists) with additional strategies to regulate these variables and expand their prescription toolkit. Therefore, the aim of the present study was to review the hemodynamic and autonomic responses to MM or FR—applied alone or in combination with ST in healthy adults.

## 2. Materials and Methods

The present review is in accordance with the Preferred Reporting Items for Systematic Reviews and Meta-Analyses for Scoping Reviews (PRISMA-ScR) guidelines [22]. The 5-step proposition outlined by Arksey and O’Malley [23] was used as guidance in the organization of the study. According to previous evidence, a scoping review can be conducted in place of a systematic review as its purpose is to identify knowledge gaps, evaluate a body of literature, clarify concepts, or investigate research behavior, without intending to indicate any clinical conduct [24].

### 2.1. Stage 1: Identification of Relevant Studies

Studies were retrieved through electronic database searches and a comprehensive scan of the reference lists of included studies. Two researchers (ERM and LMA) conducted searches in 4 databases (Nursing and Allied Health (CINAHL), Cochrane Library, PubMed^®^, and SciELO) between 20 December 2024 and 14 April 2025.

### 2.2. Stage 2: Study Selection

Study selection was conducted based on the PICOS strategy [25]: (P) a physically active population of both sexes aged between 18 and 59 years (this age range was deliberately chosen to encompass all age groups within the adult population while excluding older adults); (I) interventions of MM and FR performed in combination or separately with ST; (C) compared to a no intervention approach (control group); (O) assessing autonomic responses (heart rate variability) and hemodynamic responses (blood pressure, heart rate, double product, and arterial vascular perfusion) as outcomes; and (S) cross-sectional studies (i.e., randomized controlled or cross-over trials) were included.

The following inclusion criteria were adopted: (1) original studies published without temporal restrictions; (2) interventions based on MM and FR, along with ST; (3) studies assessing at least one of the outcomes of interest; and (4) cross-sectional study designs. The exclusion criteria were as follows: (1) duplicate studies; (2) studies not written in English or Portuguese languages; (3) studies that did not isolate or combine the effects of MM, FR, and strength training; (4) studies that tested the effects of MM, FR, and ST in populations with specific health conditions (e.g., hypertensive individuals and pregnant women); and (5) studies involving animal models.

In addition, the search strategy associated the following descriptors and Boolean operators (AND/OR/NOT): (‘myofascial release’ OR ‘self-myofascial release’ OR ‘massage’ OR ‘manual massage’ OR ‘foam rolling’ OR ‘rolling massage’) AND (‘resistance training’ OR ‘resistance exercise’ OR ‘strength training’ OR ‘strength exercise’ OR ‘weight training’ OR ‘weight exercise’ OR ‘weightlifting’ OR ‘weight-lifting’ OR ‘weight lifting’) AND (‘blood pressure’ OR ‘hemodynamic response’ OR ‘autonomic response’ OR ‘heart rate’ OR ‘heart rate variability’ OR ‘rate product pressure’ OR ‘cardiac output’ OR ‘arterial function’ OR ‘arterial tissue perfusion’ OR ‘vascular tissue perfusion’) NOT (‘review’) with its respective translation to the Portuguese.

### 2.3. Stage 3: Data Mapping

Duplicate records were systematically removed through both automated and manual processes soon after the studies were imported into EndNote X9 software (Clarivate Analytics, PA, USA). Two independent reviewers (VMR and JVA) screened the titles and abstracts based on the predefined eligibility criteria, with a third (MRA) reviewer resolving any discrepancies. Reviewers were not blinded to the authors, affiliated institutions, or publishing journals. Full-text articles were retrieved for further evaluation when abstracts did not display sufficient information for adequate analysis.

Two researchers (GSA and VGCN) extracted data from the full texts using a standardized and pre-structured protocol. The collected data included participants’ characteristics (sample size, age, height, body mass, training status, and sex) and treatment protocols (study, objective, interventions, and results). The data extracted by both researchers were compared, and any discrepancies were resolved through consensus. Whenever possible, data were directly copied and pasted to avoid any misinterpretation.

The methodological quality of the selected studies was assessed using the Centre of Evidence-Based Physiotherapy proposal [26] by two researchers (CMB and JSN). The PEDro scale comprises a list of 11 criteria. Clear and unambiguous meetings of a criterion result in the award of 1 point. Scores between 6 and 10 points, 4 and 5 points, and 0 and 3 points are classified as high, moderate, and low quality, respectively. Two authors applied the scale, and any disagreements regarding the PEDro scores classification were resolved through a consensus discussion among the authors. In cases where a consensus could not be reached, a third researcher was invited to provide their opinion. It is important to note that the PEDro scale classification was limited to describing the study quality and was not used as a criterion for study inclusion or exclusion.

### 2.4. Stage 4: Gathering, Summarizing, and Reporting the Results

A narrative description and analysis of the selected studies was conducted considering the application of a common analytical framework that made possible the adequate summarizing of the results.

## 3. Results

Figure 1 summarizes the outcome of each stage of the research.

After adequate application of both inclusion and exclusion criteria, seven studies were selected for the current review. The details of the characteristics of the 95 participants and of the included studies are presented in Table 1 and Table 2, respectively.

### 3.1. Study Quality

Scores attained by each investigation according to the PEDro scale are depicted in Table 3, with the studies presenting a moderate to high rating. Therefore, there was no substantive variation in the quality among the selected studies.

### 3.2. FR Alone

Five included studies point to an improvement in hemodynamic responses post-FR performed alone, such as a reduction in arterial stiffness [12] and systolic BP [27,28], as well as increased vasodilatory responses due to higher nitric oxide concentration (from 20.4 ± 6.9 to 34.4 ± 17.2 μmol/L) [12] and blood flow between elbow and ankle (from 1202 ± 105 to 1074 ± 110 cm/s) [12]. In this way, Hotfiel et al. [11] observed increases in Doppler peak flow in 0 and 30 post-FR (73.6% and 52.7% respectively), in Doppler time average velocity maximum in 0 and 30 post-FR (53.2% and 38.3%), and Doppler time average velocity mean in 0 and 30 post-FR (84.4% and 68.2%, respectively). Finally, Ketelhut et al. [30] observed a PEH effect on diastolic BP from post–pre to the control condition (*p* < 0.001), resulting in a significantly lower value at post (*p* = 0.027) compared to post–pre following the self-myofascial release by FR (*p* < 0.001), and resulting in a significantly lower value at post (*p* = 0.030). Ketelhut et al. [30] observed a reduction in total peripheral resistance from post–pre (*p* = 0.017), resulting in a significantly lower value at post (*p* = 0.024).

Only two studies [27,30] explored the effects of heart rate variability after the isolated application of FR. The authors showed improved autonomic response, tending towards reduced sympathetic activity [27]. Ketelhut et al. [30] observed a decrease in heart rate (*p* = 0.043) from post–pre (*p* = 0.017), leading to a lower value at t1 (*p* < 0.001) and t2 (*p* = 0.007).

### 3.3. MM Alone

Only a single study investigated the isolated effects of MM alone on PEH [21]. Monteiro et al. [21] reported a reduction in systolic BP of −4 mmHg at Post-50 (*p* = 0.011; d = −2.61) and Post-60 (*p* = 0.011; d = −2.74).

### 3.4. Combined Interventions

None of the studies included in this scoping review reported significant PEH following the combined interventions of ST + FR or ST + MM.

## 4. Discussion

This scoping review highlights the acute effects of MM and FR, with or without ST, on hemodynamic and autonomic responses in healthy adults. The findings indicate that these interventions can elicit immediate physiological changes, which may support their application as short-term strategies for modulating cardiovascular and autonomic function in normotensive individuals [11,12,21,27,28,29,30]. These insights may be particularly valuable for health and exercise professionals seeking to optimize recovery, cardiovascular regulation, or performance. However, given that all included studies focused on acute responses, any assumptions regarding long-term outcomes remain premature. Therefore, well-designed longitudinal studies are essential to determine the sustained impact of these interventions and to establish evidence-based recommendations for their integration into long-term health or performance programs.

Walaszek [31] conducted a study involving ten massage therapy sessions targeting the lower limbs of elderly hypertensive women, reporting a significant PEH in SBP. Similarly, Givi et al. [32] observed PEH in SBP through 50 pre-hypertensive women following Swedish Therapeutic Massage with a non-aromatic topical lotion applied to the face, neck, shoulders, and upper chest. This intervention, performed three times per week for 10–15 min per session over 3.5 weeks, utilized both superficial and deep strokes in the supine position. Additionally, a systematic review with meta-analysis by Liao et al. [13] demonstrated that various massage therapy techniques—including Swedish Therapeutic Massage, chair massage, light touch massage, and soothing touch massage—significantly affected PEH in SBP (−7.39 mmHg; SE = −0.728) and DBP (−5.04 mmHg; SE = −0.334). These findings collectively reinforce the primary observations of the present review. However, the aforementioned studies were not included due to methodological differences in the techniques assessed. The convergence of findings across different massage approaches suggests a shared underlying mechanism, likely linked to therapeutic touch [33,34,35,36] and its capacity to elicit central nervous system responses. White and Raven [37] highlight that, during exercise, autonomic nervous system regulation shifts to maintain homeostasis, leading to a reduction in vagal activity. It is postulated that mechanoreceptors within the muscle and fascia, upon activation, diminish muscle tone, enhance parasympathetic activity, and promote the release of neuropeptides and endocannabinoids, ultimately contributing to blood pressure reductions [14].

Despite the variability in techniques, both MM and stretching interventions exhibit a significant PEH, emphasizing the role of touch perception in these strategies. Inami et al. [20] reported that while SBP temporarily increased during static stretching, it promptly returned to baseline post-intervention. Similarly, Da Silva Araújo et al. [38] found a 6.1% SBP reduction following isolated static stretching. Furthermore, they explored the effects of combining ST with SS and whether the sequence of execution influenced BP responses. Their findings revealed a significant HPE response regardless of order. Souza et al. [39] corroborated these results, noting SBP reductions of up to 12.2 mmHg within 60 min post-ST + SS. A plausible mechanism underlying these effects involves static stretching-induced vascular changes, where mechanical compression from muscle elongation transiently narrows blood vessels, thereby modulating blood flow and nutrient delivery. Kruse and Scheuermann [40] described how mechanical vascular deformation during the initial phase of stretching, alongside stimulation of group III afferent fibers, triggers a cascade of physiological responses—peripheral vasodilation, increased heart rate, elevated cardiac output, and subsequent BP modulation.

Our findings highlight favorable cardiovascular adaptations, even when massage interventions—whether MM or FR—were applied in isolation. Lastova et al. [27] analyzed BP responses at 10 and 30 min post-FR targeting the thigh (adductors, posterior, anterior, and lateral), calf (gastrocnemius), and back (upper and lower). Their results revealed a significant PEH in SBP reduction alongside increased vagal modulation lasting up to 30 min post-intervention. These outcomes align with Monteiro et al. [28], who reported substantial PEH in SBP (effect size d = −0.98 to −3.26) following FR, reinforcing the observations of Lastova et al. [27]. Collectively, these findings suggest that FR, whether administered independently or in conjunction with ST, holds potential clinical relevance, particularly for individuals with limited access to regular exercise. Finally, it is important to highlight that Lastova et al. [27] assessed BP only up to 30 min post-FR, leaving an open question as to whether these reductions persist beyond this timeframe. The precise physiological mechanisms underlying BP modulation following FR remain under investigation. Okamoto et al. [12] reported elevated nitric oxide concentrations post-FR, suggesting an enhanced vasodilatory response that could contribute to reductions in SBP, double product, and heart rate. Similarly, Hotfiel et al. [11] observed increased local arterial perfusion in the lateral thigh region after FR, attributing these hemodynamic changes to nitric oxide-mediated vasodilation induced by the intervention.

Limitations are intrinsic to the scientific method. Still, the authors highlight that the review was conducted in accordance with well-established parameters such as guidelines of the Preferred Reporting Items for Systematic Reviews and Meta-Analyses for Scoping Reviews (PRISMA-ScR) [22], aiming to allow for reproducibility. The search was conducted in four major electronic databases (Nursing and Allied Health (CINAHL), Cochrane Library, PubMed^®^, and SciELO), excluding other databases and possible “gray literature”. This conduct may have excluded relevant additional studies. Still, only seven articles were included in this review. While this number may be seen as a potential limitation, it represents the state-of-the-art literature in the searched databases. The low number of retrieved articles may also be a result of excluding preprint articles due to lack of an adequate peer review process. Furthermore, the methodology of this review was prospectively registered on the Open Science Framework (doi: osf.io/kfcr5), and the full article was also made publicly available as a preprint (doi: 10.20944/preprints202502.1932.v1). Another limitation is the inclusion of only cross-sectional studies, primarily evaluating acute responses to the interventions (i.e., results immediately post-intervention, from Post-0 to Post-60). This focus on short-term effects limits the external validity of the findings, as the results provide only transient insights into hemodynamic and autonomic changes. Consequently, conclusions about long-term or chronic effects cannot be drawn, restricting the ability to assess sustained impacts or potential adaptations resulting from repeated or prolonged exposure to these interventions. Future research should incorporate longitudinal designs to better understand the cumulative and lasting effects of massage techniques and their combination with strength training. Although all articles included in the results of this review demonstrate moderate to high methodological quality, the lack of blinding in most studies is an important concern, as shown in Table 3. The absence of participant and/or assessor blinding increases the risk of performance and detection bias, especially in studies evaluating subjective or autonomic outcomes, which may be influenced by participants’ expectations or the experimenter’s presence. While we acknowledge that implementing blinding in interventions such as massage or foam rolling is inherently challenging, this limitation compromises the internal validity and reliability of the results. Future studies should aim to minimize this bias by using alternative strategies, such as blinded outcome assessors or employing objective measurement tools whenever possible. Finally, the limited sample size in some studies, as well as the variability in participant characteristics (e.g., age and fitness level), further restricts the generalizability of the findings. Larger, more diverse samples would provide more robust data and allow for a better understanding of how these interventions may affect different populations.

## 5. Conclusions

There are positive acute responses in hemodynamic and autonomic variables, which may help inform decision-making for professionals prescribing exercise (e.g., Exercise Physiologists, Biomechanics, and Physical Therapists) to influence hemodynamic and autonomic responses in normotensive individuals. Although the investigations included in this review were acute, the observed data suggest that both MM and FR can be powerful tools for improving hemodynamic and autonomic aspects. However, it is emphasized that long-term and follow-up studies are essential for a better understanding of these parameters.

## Figures and Tables

**Figure 1 healthcare-13-01371-f001:**
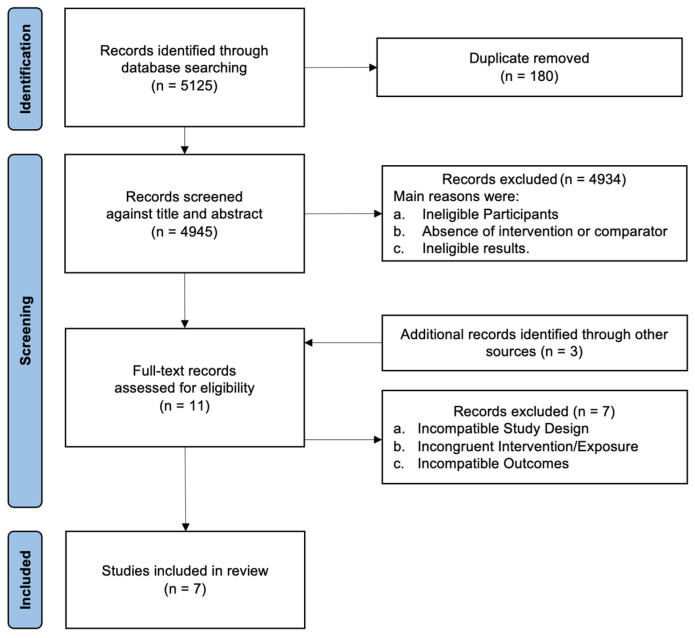
Flowchart.

**Table 1 healthcare-13-01371-t001:** Characteristics of the participants (n = 7 studies).

Studies	Sample Size (n = 95)	Sex	Age (Years)	Height	Body Mass (kg)	Training Status
Okamoto et al. [12]	10 (7 men and 3 women)	Both sexes	19.9 ± 0.3	162.7 ± 8.1 cm	60.6 ± 11.2	Recreational strength training
Hotfiel et al. [11]	21 (12 men and 9 women)	Both sexes	25 ± 2	177 ± 9 cm	74 ± 9	Recreational strength training
Lastova et al. [27]	15 (8 men and 7 women)	Both sexes	21.55 ± 0.52	1.72 ± 0.02 m	74.79 ± 2.88	N/A
Monteiro et al. [21]	16	Women	25.1 ± 2.9	158.9 ± 4.1 cm	59.5 ± 4.9	Recreational strength training
Monteiro et al. [28]	16	Women	25.5 ± 2.0	155.7 ± 4.4 cm	61.2 ± 5.4	Recreational strength training
Monteiro et al. [29]	12	Women	27.2 ± 3.3	164.8 ± 5.5 cm	69.8 ± 6.0	Recreational strength training
Ketelhut et al. [30]	20	Men	26 ± 2	182.6 ± 6.9	76.1 ± 7.2	N/A

N/A = not applicable.

**Table 2 healthcare-13-01371-t002:** Summary and characteristics of the studies included in the review (n = 7 studies).

Studies	Objective	Interventions		Results
Okamoto et al. [12]	To investigate the acute effect of FR on arterial stiffness and vascular endothelial function.	FR	FR condition was performed on the adductors, hamstrings, quadriceps, iliotibial band, and trapezius regions. Each participant practiced 2 or 3 times to learn the correct FR technique with the guidance of a coach and performed 20 repetitions in each region with 1 min intervals. The pressure was self-adjusted by applying body weight to the roller and using hands and feet to regulate pressure as needed. The roller was placed under the target tissue area, and the body was moved back and forth along the roller.	↓ Ankle-brachial PWV (from 1202 ± 105 to 1074 ± 110 cm/s). ↑ Plasma nitric oxide concentration (from 20.4 ± 6.9 to 34.4 ± 17.2 μmol/L). Both after FR (*p* < 0.05), but neither of them differed significantly after the control condition.
		CON	Rest in lying position in a quiet temperature-controlled room.	
Hotfiel et al. [11]	To evaluate the effect of FR on arterial blood flow in the lateral thigh region.	ST + FR	The exercise protocol consisted of 3 sets, each with 45 s of FR on the lateral thigh region in the sagittal plane (with 20 s of rest between sets).	↑ Arterial blood flow of the lateral thigh increased significantly after FR exercises compared with baseline (*p* ≤ 0.05). ↑ Vmax of 73.6% (0 min) and 52.7% (30 min) (*p* < 0.001), in TAMx of 53.2% (*p* < 0.001) and 38.3% (*p* = 0.002), and in TAMn of 84.4% (*p* < 0.001) and 68.2% (*p* < 0.001).
Lastova et al. [27]	To assess the effects of an acute FR session on HRV and BP in healthy individuals.	FR	In the FR condition, individuals completed 10 repetitions of FR per target area of the body (adductors, hamstrings, quadriceps, iliotibial band, gastrocnemius, and upper trapezius), followed by 1 min of rest. Each repetition involved moving the target tissue across the roller in a smooth motion at a rate of 2 s down and 2 s up, as determined by a metronome.	↑ in vagal tone index (normalized high-frequency power) 30-min after FR (*p* < 0.01), while no changes were observed after the control condition. ↓ sympathetic activity (*p* < 0.05) (normalized low-frequency power) and sympathovagal balance (normalized low-frequency to high-frequency ratio).
		CON	The control condition only involved measurements without the application of other experimental conditions.	
Monteiro et al. [21]	To examine the acute effects of resistance exercise and different manual therapies (SS and MM) performed separately or combined on BP responses during recovery in normotensive women.	MM	The isolated SS and isolated MM conditions were applied unilaterally in two sets of 120 s for each quadriceps, hamstrings, and calf region.	↓ Systolic BP in isolated strength training at Post-50 (*p* = 0.038; d = −2.24; ∆ = −4.0 mmHg), isolated SS at Post−50 (*p* = 0.021; d = −2.67; ∆ = −5.0 mmHg), and Post-60 (*p* = 0.008; d = −2.88; ∆ = −5.0 mmHg), and isolated MM at Post-50 (*p* = 0.011; d = −2.61; ∆ = −4.0 mmHg) and Post-60 (*p* = 0.011; d = −2.74; ∆ = −4.0 mmHg). ↓ Systolic BP in the combined of strength training and SS at Post-60 (*p* = 0.024; d = −3.12; ∆ = −5.0 mmHg).
		SS	The isolated SS and isolated MM conditions were applied unilaterally in two sets of 120 s for each quadriceps, hamstrings, and calf region.	
		ST	Isolated strength training comprised three sets of bench press, back squat, and leg press at an intensity controlled to 80% of 10RM.	
		ST + MM	In the combined condition of strength training and MM, the massage was conducted immediately after strength training, following the same descriptions as above.	
		ST + SS	In the combined condition of strength training and SS, SS was performed immediately after strength training, following the same descriptions as above.	
		CON	The control condition consisted solely of measurements without applying any other experimental conditions.	
Monteiro et al. [28]	Toexamine the acute effects of resistance exercise and FR performed separately or combined on BP responses during recovery in normotensive women.	FR	In the isolated FR condition, foam rolling was performed unilaterally in two sets of 120 s for each quadriceps, hamstrings, and calf region.	↓ Systolic BP in isolated strength training at Post-50 (*p* < 0.001; d = −2.14) and Post-60 (*p* = 0.008; d = −2.88), and in isolated FR at Post-60 (*p* = 0.020; d = −2.14). ↓ Systolic BP in the combined condition of strength training and FR at Post-50 (*p* = 0.001; d = −2.03) and Post-60 (*p* < 0.001; d = −2.38).
		ST	Isolated strength training comprised three sets of bench press, back squat, lateral pulldown, and leg press at an intensity controlled to 80% of 10RM.	
		ST + FR	In the combined condition of strength training and FR, FR was conducted immediately after strength training, following the same descriptions as above.	
		CON	The control condition consisted solely of measurements without applying any other experimental conditions.	
Monteiro et al. [29]	To examine the acute effects of different pre-strength training strategies on total training volume, maximum repetition performance, fatigue index, and blood pressure responses in recreationally strength-trained women.		10RM test and retest for bench press 45°, front squat, lat pulldown, leg press, shoulder press, and leg extension. Strength Training = 80% of 10RM load with self-suggested rest interval. FR and SS = Applied, unilaterally, in randomized order, in single set of 90 s to the lateral torso of the trunk, anterior and posterior thigh, and calf regions. Aerobic Exercise = Walking on the treadmill with intensity between 30% and 60% of the heart rate reserve. Specific Warm-Up = Two sets of 15 repetitions with 40%10RM with 90 s rest interval. BP was measured at baseline, Post-10, Post-20, Post-30, Post-40, Post-50, and Post-60 min.	No significant reductions were observed for systolic and diastolic BP with effect sizes magnitude ranging between trivial and large.
Ketelhut et al. [30]	To evaluate the immediate effects of acute self-myofascial release on peripheral and central BP, HR, HRV, TPR, and PWV. Investigate whether self-myofascial release can influence hemodynamic reactivity and perceived pain during a cold pressor test.		FR Self-myofascial release was performed by two sets of 60 s with a 60 s rest interval between sets, targeting the calf, outer thigh, front thigh, inner thigh, and buttocks muscles regions. Hemodynamic and cardiac autonomic parameters were evaluated at rest (t0) and during a cold pressor test (CPT_t0). Following this, participants either engaged in a 20 min SMR exercise or a 20 min seated rest (CON). After each condition, outcomes were assessed again at rest (t1) and during a cold pressor test 2 min after the condition (CPT_t1), as well as after a 20 min period of supine rest (t2, CPT_t2).	↓ Diastolic BP (*p* < 0.001) from t0 to t2 compared to the control condition (*p* < 0.001), resulting in a significantly lower at t2 (*p* = 0.027). ↓ Diastolic BP (*p* < 0.001) from t0 to t2 following the self-myofascial release (*p* < 0.001). This resulted in a significantly lower at t2 (*p* = 0.030). ↓ TPR from t0 to t2 (*p* = 0.017), resulting in a significantly lower TPR at t2 (*p* = 0.024). No time × condition interaction effects could be observed for systolic BP, PWV, and HRV (LF/HF parameters) (*p* > 0.05). ↓ HR (*p* = 0.043) from t0 to t1 (*p* 0.017), leading to a lower HR at t1 (*p* < 0.001) and t2 (*p* = 0.007). ↑ HRV (RMSSD index) from t0 to t1 was detected following the control condition (*p* = 0.047), leading to a significantly lower value after the control condition at t1 (*p* = 0.006).

BP = blood pressure; FR = foam rolling; HR = heart rate; HRV = heart rate variability; MM = manual massage; SS = static stretching; PWV = pulse wave velocity; TAMx = time average velocity maximum; TAMn = time average velocity mean; TPR = total peripheral resistance; Vmax = peak flow. ↓ = decreased; ↑ = increased.

**Table 3 healthcare-13-01371-t003:** Methodological quality information. of the studies included in the review (n = 7 studies).

Studies	Eligibility Criteria	Random Allocation	Concealed Allocation	Baseline Comparability	Blind Subjects	Blind Therapists	Blind Assessors	Adequate Follow-Up	Intention-to-Treat Analysis	Between-Group Comparisons	Point Estimates and Variability
Okamoto et al. [12]	No	Yes	No	Yes	No	No	No	Yes	No	Yes	Yes
Hotfiel et al. [11]	No	Yes	No	Yes	No	No	No	Yes	No	Yes	Yes
Lastova et al. [27]	Yes	Yes	Yes	Yes	No	No	No	Yes	No	Yes	Yes
Monteiro et al. [21]	Yes	Yes	Yes	Yes	No	No	No	Yes	No	Yes	Yes
Monteiro et al. [28]	Yes	Yes	Yes	Yes	No	No	No	Yes	No	Yes	Yes
Monteiro et al. [29]	Yes	Yes	Yes	Yes	No	No	No	Yes	No	Yes	Yes
Ketelhut et al. [30]	Yes	Yes	Yes	Yes	Yes	No	No	Yes	No	Yes	Yes

The studies included in this review were grouped according to the type of intervention: FR alone, MM alone, and combined interventions. Among studies involving FR alone, reductions were observed in arterial stiffness, systolic BP, sympathetic activity, and heart rate, along with increases in vasodilation and blood flow velocity. For studies involving MM alone, the cardiovascular response was characterized by PEH in systolic BP. Lastly, combined interventions appear to enhance PEH and report reductions in systolic BP.

## Data Availability

Data sharing is not applicable. No new data were created or analyzed in this study.

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
