# Peer review of "Effect of Manual Massage, Foam Rolling, and Strength Training on Hemodynamic and Autonomic Responses in Adults: A Scoping Review"

_healthcare, 2025, doi:10.3390/healthcare13121371_

Round 1
Reviewer 1 Report (Previous Reviewer 1)
Comments and Suggestions for Authors
- Registering a systematic review on these platforms ensures transparency and avoids duplication of effort, serving as a prerequisite for publication. A preprint, however, involves sharing the full text before formal publication. Both are useful but serve different purposes.
We appreciate the suggestion and, although it was not feasible to implement it in the current manuscript, it will certainly be taken into consideration for future studies.
RESPONSE: I believe that in the present study it can be done.
- Figure 1. Flow diagram. Improve the quality of the image because it is not displayed well. It does not correctly represent the article selection process. Duplicates are eliminated first. The reasons for exclusion based on Title or Abstract are not understood. There are numerical errors. Is the final sample 5 or 6 studies?
We have now updated the database search and made a corresponding revision to Figure 1.
RESPONSE: I keep seeing errors. If there are 4934 documents in the screening phase, why are there only 11 in the eligibility phase?. It would be necessary to specify what happened to the missing documents and explain what happened to the 3 documents included from other sources.
Author Response
- Registering a systematic review on these platforms ensures transparency and avoids duplication of effort, serving as a prerequisite for publication. A preprint, however, involves sharing the full text before formal publication. Both are useful but serve different purposes.
Authors Response: We appreciate the suggestion and, although it was not feasible to implement it in the current manuscript, it will certainly be taken into consideration for future studies.
Reviewer Response: I believe that in the present study it can be done.
Authors Response: Thank you for highlighting the importance of preregistration. Initially, this was not implemented due to timing constraints. However, in response to your feedback, we have now registered the scoping review protocol on the Open Science Framework (OSF), which enhances the transparency and reproducibility of our work. The registration is available at the following link: Monteiro, E. R., PhD. (2025, April 28). Effect of Manual Massage, Foam Rolling, and Strength Training on Hemodynamic and Autonomic Responses in Adults: A Scoping Review. OSF. https://doi.org/10.17605/OSF.IO/HWXUP. Additionally, we have acknowledged the absence of preregistration as a limitation and discussed this point in the revised Discussion section.
- Figure 1. Flow diagram. Improve the quality of the image because it is not displayed well. It does not correctly represent the article selection process. Duplicates are eliminated first. The reasons for exclusion based on Title or Abstract are not understood. There are numerical errors. Is the final sample 5 or 6 studies?
Authors Response: We have now updated the database search and made a corresponding revision to Figure 1.
Reviewer Response: I keep seeing errors. If there are 4934 documents in the screening phase, why are there only 11 in the eligibility phase?. It would be necessary to specify what happened to the missing documents and explain what happened to the 3 documents included from other sources.
Authors Response: We corrected the numerical inconsistencies and ensured that all transitions between phases are clearly documented and justified. The updated figure is of higher resolution and is now embedded in the revised manuscript. The final sample consists of 7 studies, as now clearly indicated.
Reviewer 2 Report (New Reviewer)
Comments and Suggestions for Authors
The manuscript presents a timely and relevant scoping review that evaluates the acute effects of manual massage (MM), foam rolling (FR), and strength training (ST)—both isolated and in combination—on hemodynamic and autonomic responses in healthy adults. The topic is important given the growing interest in non-pharmacological interventions for cardiovascular modulation. The manuscript is generally well-structured and clear, but it contains several methodological, interpretative, and reporting issues that should be addressed to enhance its scientific quality and clarity.
Major Comments
Justification for Scoping Review Approach:
While the authors state adherence to PRISMA-ScR and Arksey and O'Malley’s framework, the rationale for choosing a scoping review over a systematic review remains underdeveloped. Given the limited number of studies included (n = 7), a systematic review could have been feasible. Please justify why a scoping approach was used instead of a more rigorous systematic review with meta-analysis.
Lack of Clarity in Objectives:
The aim is repeated multiple times, but phrased inconsistently. For example, lines 34–36 and 97–99 essentially repeat the same purpose. Please clearly and concisely state the primary objective once, ideally at the end of the introduction.
Methodological Weaknesses:
The inclusion of only cross-sectional designs (e.g., acute interventions) is a major limitation. Yet this is not critically discussed in the main text.
The absence of blinding in most studies, acknowledged in Table 3, raises concerns about potential bias.
Expand the discussion of methodological limitations and clearly describe how these limitations impact the generalizability and strength of the findings.
Synthesis of Results:
The results are overly descriptive and lack critical synthesis. For instance, reporting exact p-values without context or grouping studies by intervention type weakens interpretability. As a recommendation; Group studies by intervention type (e.g., FR only, MM only, combined with ST), and compare their findings systematically.
Terminological and Conceptual Inconsistency:
Terms like “autonomic response” and “sympathovagal control” are used interchangeably without proper explanation or definition. Provide clear definitions of key physiological terms and maintain consistency in their usage throughout the manuscript.
Interpretation of Evidence:
There is an overinterpretation of the acute responses. For instance, suggesting "powerful tools" for long-term health benefits (line 309) based solely on acute data is speculative. Temper conclusions to reflect the limitations of acute studies and call for longitudinal evidence.
Minor Comments
Language and Style:
The manuscript contains grammatical and typographical issues (e.g., “There is a positive acutely responses” in line 305). A thorough language revision is necessary. Consider professional English editing.
PRISMA Flow Diagram:
Figure 1 is mentioned but not shown in the extracted text. Ensure the PRISMA-ScR flowchart is detailed and adheres to standard formatting.
Data Extraction Table:
Tables 1 and 2 are informative but dense. Use shading or grouping to visually separate intervention types and outcomes for easier readability.
Reporting Standards:
The manuscript should include a registered protocol ID (e.g., Open Science Framework) if available, or explicitly state that it was not preregistered. Add a registration statement to improve transparency.
References:
The references are adequate but dominated by the authors’ own previous works. This raises a concern about potential selection bias. Broaden the literature base and include more independent sources, particularly meta-analyses or large-scale trials.

The manuscript contains grammatical and typographical issues (e.g., “There is a positive acutely responses” in line 305). A thorough language revision is necessary. Consider professional English editing.
Author Response
The manuscript presents a timely and relevant scoping review that evaluates the acute effects of manual massage (MM), foam rolling (FR), and strength training (ST)—both isolated and in combination—on hemodynamic and autonomic responses in healthy adults. The topic is important given the growing interest in non-pharmacological interventions for cardiovascular modulation. The manuscript is generally well-structured and clear, but it contains several methodological, interpretative, and reporting issues that should be addressed to enhance its scientific quality and clarity.
Authors Response: We appreciate it and have revised the manuscript as requested.
Major Comments
- Justification for Scoping Review Approach: While the authors state adherence to PRISMA-ScR and Arksey and O'Malley’s framework, the rationale for choosing a scoping review over a systematic review remains underdeveloped. Given the limited number of studies included (n = 7), a systematic review could have been feasible. Please justify why a scoping approach was used instead of a more rigorous systematic review with meta-analysis.
Authors Response: We appreciate the reviewer’s comment regarding our choice of methodological framework. Importantly, the decision to conduct a scoping review was made a priori, during the study planning phase—before performing the search or knowing how many studies would ultimately meet the inclusion criteria. Our primary aim was to map the breadth of existing evidence, characterize the interventions, and identify knowledge gaps related to the acute effects of manual massage (MM), foam rolling (FR), and their combination with strength training (ST) on hemodynamic and autonomic responses. Given the emerging and heterogeneous nature of this field—with considerable variability in study designs, participant characteristics, intervention protocols, and outcome measures—a scoping review was considered the most suitable approach. As outlined by Arksey and O'Malley [23] and supported by Munn et al. [24], scoping reviews are particularly appropriate when the research field is still developing, and when the goal is to provide an overview rather than synthesize data quantitatively. While the final number of included studies (n = 7) may seem small, this was not known in advance, and the diversity of the studies further limits the feasibility of a meta-analysis at this stage. We have added this clarification to the Introduction section of the revised manuscript.
- Lack of Clarity in Objectives: The aim is repeated multiple times, but phrased inconsistently. For example, lines 34–36 and 97–99 essentially repeat the same purpose. Please clearly and concisely state the primary objective once, ideally at the end of the introduction.
Authors Response: We appreciate the reviewer’s observation regarding the repetition and inconsistency in the phrasing of the study’s objective. To address this, we have revised the manuscript to state the primary objective only once, clearly and concisely, at the end of the introduction. We have also reformulated the abstract accordingly.
- Methodological Weaknesses: The inclusion of only cross-sectional designs (e.g., acute interventions) is a major limitation. Yet this is not critically discussed in the main text.
Authors Response: We thank the reviewer for this important observation. We agree that the exclusive inclusion of cross-sectional studies, primarily assessing acute interventions, constitutes a methodological limitation. In response, we have added a paragraph to the Discussion section explicitly acknowledging and addressing this issue. We also emphasize the need for future studies employing longitudinal or repeated-intervention designs to assess long-term effects and adaptations.
‘Another limitation is the inclusion of only cross-sectional studies, primarily evaluating acute responses to the interventions (i.e., results immediately post-intervention, from Post-0 to Post-60). This focus on short-term effects limits the external validity of the findings, as the results provide only transient insights into hemodynamic and autonomic changes. As a result, it is not possible to draw conclusions about long-term or chronic effects, nor to evaluate sustained adaptations resulting from repeated or prolonged exposure to MM, FR, or their combination with ST. Future studies should incorporate longitudinal designs to better understand the cumulative and lasting effects of massage techniques and their combination with strength training.’
- The absence of blinding in most studies, acknowledged in Table 3, raises concerns about potential bias.
Authors Response: We appreciate the reviewer’s comment regarding the lack of blinding in most of the included studies. While this issue was already noted in Table 3, we agree that it should be explicitly addressed in the main text. We have now added a statement in the discussion section highlighting the potential for performance and detection bias due to the absence of participant and/or assessor blinding, and how this may affect the internal validity of the results.
‘Although all articles included in the results of this review demonstrate moderate to high methodological quality, the lack of blinding in most studies is an important concern, as shown in Table 3. The absence of participant and/or assessor blinding increases the risk of performance and detection bias, especially in studies evaluating subjective or autonomic outcomes, which may be influenced by participants' expectations or the experimenter's presence. While we acknowledge that implementing blinding in interventions such as massage or foam rolling is inherently challenging, this limitation compromises the internal validity and reliability of the results. Future studies should aim to minimize this bias by using alternative strategies, such as blinded outcome assessors or employing objective measurement tools whenever possible.’
- Expand the discussion of methodological limitations and clearly describe how these limitations impact the generalizability and strength of the findings.
Authors Response: We appreciate the reviewer’s valuable comment. In response, we have expanded the Discussion section to explicitly address several methodological limitations of the included studies. These include the use of only cross-sectional designs, small and homogeneous sample sizes, limited blinding, and variability in intervention protocols and outcome measures. We have also discussed how these factors affect both the internal validity and generalizability of our findings. Specific changes have been made in the paragraph discussing limitations.
- Synthesis of Results: The results are overly descriptive and lack critical synthesis. For instance, reporting exact p-values without context or grouping studies by intervention type weakens interpretability. As a recommendation; Group studies by intervention type (e.g., FR only, MM only, combined with ST), and compare their findings systematically.
Authors Response: We appreciate the reviewer’s suggestion regarding the synthesis of results. We agree that the presentation of p-values in isolation may limit the interpretability of the findings. In response to this feedback, we have reorganized the results section to group studies systematically by intervention type (e.g., Foam Rolling (FR) only, Manual Massage (MM) only, and combined with Strength Training (ST)). This restructuring allows for a clearer comparison of findings within each intervention type, providing a more critical analysis of the effects observed. Additionally, we have included a more in-depth discussion of the p-values, incorporating the context of the studies and their implications for the field. This approach aims to enhance the overall synthesis and make the results more interpretable and meaningful.
- Terminological and Conceptual Inconsistency: Terms like “autonomic response” and “sympathovagal control” are used interchangeably without proper explanation or definition. Provide clear definitions of key physiological terms and maintain consistency in their usage throughout the manuscript.
Authors Response: We appreciate the comment. We have now standardized the terminology, using only the term “autonomic response.” The variable representing the autonomic response evaluated in this scoping review—heart rate variability—is presented in lines 120–121.
- Interpretation of Evidence: There is an overinterpretation of the acute responses. For instance, suggesting "powerful tools" for long-term health benefits (line 309) based solely on acute data is speculative. Temper conclusions to reflect the limitations of acute studies and call for longitudinal evidence.
Authors Response: We thank the reviewer for the valuable comment regarding the interpretation of acute findings. We agree that the previous wording overemphasized the potential long-term impact of the interventions based solely on acute data. In response, we have revised the discussion and conclusion to avoid speculative language and to better reflect the scope of the evidence. The updated paragraph now highlights the short-term nature of the findings and includes a clear call for longitudinal studies to explore potential long-term effects.
This scoping review highlights the acute effects of MM and FR, with or without ST), on hemodynamic and autonomic responses in healthy adults. The findings indicate that these interventions can elicit immediate physiological changes, which may support their application as short-term strategies for modulating cardiovascular and autonomic function in normotensive individuals [11,12,21,27,28,29,30]. These insights may be particularly valuable for health and exercise professionals seeking to optimize recovery, cardiovascular regulation, or performance. However, given that all included studies focused on acute responses, any assumptions regarding long-term outcomes remain premature. Therefore, well-designed longitudinal studies are essential to determine the sustained impact of these interventions and to establish evidence-based recommendations for their integration into long-term health or performance programs.
Minor Comments
- Language and Style: The manuscript contains grammatical and typographical issues (e.g., “There is a positive acutely responses” in line 305). A thorough language revision is necessary. Consider professional English editing.
Authors Response: We appreciate the reviewer’s comment regarding the language and style of the manuscript. We acknowledge the presence of grammatical and typographical errors, including the specific example mentioned in line 305. In response, we have thoroughly revised the manuscript to improve the clarity, grammar, and overall language quality. All sentences have been reviewed and rewritten where necessary to ensure proper syntax and academic tone.
- PRISMA Flow Diagram: Figure 1 is mentioned but not shown in the extracted text. Ensure the PRISMA-ScR flowchart is detailed and adheres to standard formatting.
Authors Response: We thank the reviewer for pointing out the omission of the PRISMA-ScR flow diagram (Figure 1). We have now ensured that the flowchart is included in the revised manuscript.
- Data Extraction Table: Tables 1 and 2 are informative but dense. Use shading or grouping to visually separate intervention types and outcomes for easier readability.
Authors Response: We appreciate the reviewer’s suggestion to improve the readability of Tables 1 and 2. In response, we have reformatted both tables to enhance visual clarity.
- Reporting Standards: The manuscript should include a registered protocol ID (e.g., Open Science Framework) if available, or explicitly state that it was not preregistered. Add a registration statement to improve transparency.
Authors Response: We have now submitted our scoping review protocol through the OSF, which is available at the link below:
The registration is available at the following link: Monteiro, E. R., PhD. (2025, April 28). Effect of Manual Massage, Foam Rolling, and Strength Training on Hemodynamic and Autonomic Responses in Adults: A Scoping Review. OSF. https://doi.org/10.17605/OSF.IO/HWXUP.
- References: The references are adequate but dominated by the authors’ own previous works. This raises a concern about potential selection bias. Broaden the literature base and include more independent sources, particularly meta-analyses or large-scale trials.
Authors Response: We appreciate the reviewer’s observation. We recognize the importance of a balanced and diverse reference list to reduce the perception of selection bias. We would like to emphasize that the current scoping review was conducted following a predefined protocol and inclusion criteria, which led to the selection of seven eligible studies — three of which were authored by members of our group. This proportion reflects the current composition of the evidence base rather than a preferential selection. The field of research on the acute hemodynamic and autonomic effects of manual massage and foam rolling is still relatively narrow, with limited groups contributing original studies on these specific interventions and outcomes. As such, the inclusion of our own work is a result of the systematic search strategy and is representative of the available peer-reviewed literature at the time of review. We agree that broader representation will be critical as the field matures, and we hope this review will stimulate further independent investigations to expand the evidence base.
This manuscript is a resubmission of an earlier submission. The following is a list of the peer review reports and author responses from that submission.
Round 1
Reviewer 1 Report
Comments and Suggestions for Authors
Thank you for the opportunity to review this work that examines the cardiovascular responses after experimental conditions (Manual Massage or Foam Rolling) combined or not with Strength.
Below I have listed aspects to improve:
KEYWORDS
It is recommended to use terms that appear in a thesaurus. This term “exercise cardiology” does not appear as such. It is recommended to replace “myofascial release” with the MESH term “Myofascial Release Therapy”.
INTRODUCTION
-“These findings support a similar responsive hypothesis for the manual massage (MM) technique given the similarity in the application of both techniques (FR vs. MM – pressure and tissue sliding)”. This statement is not adequate. Massage techniques do not consist only of pressure and sliding on the tissue. They are more complex and varied maneuvers than those applied with Foam Rolling. Argue it in a better way.
On the other hand, it is recommended to provide more evidence on the mechanisms of manual massage on blood pressure.
MATERIALS AND METHODS
-Stage 2: Identification of Relevant Studies: Was the search conducted on a single day, January 29, 2025? Who reproduced it, one or more people?
-Stage 3: Study Selection: justify this inclusion criterion “aged between 18 and 59 years”.
-It is striking that only randomized controlled or counterbalanced crossover designs were included in the selection of studies. A scoping review should not be restrictive and should also include other study designs.
-Although scoping reviews cannot be registered in PROSPERO, it is recommended that the review protocol be freely available through platforms such as Figshare, Open Science Framework, ResearchGate, Research Square.
RESULTS
-Results on cardiac output, and arterial vascular perfusion are missing.
-Make it clear whether 2 or 3 independent reviewers were involved in the selection, data extraction and methodological quality assessment phases.
-Figure 1. Flow diagram. Improve the quality of the image because it is not displayed well. It does not correctly represent the article selection process. Duplicates are eliminated first. The reasons for exclusion based on Title or Abstract are not understood. There are numerical errors. Is the final sample 5 or 6 studies?.
-Table 1. Characteristics of the participants (n = 6 studies). Describe better the characteristics of the studies that are part of the review. Data from the Sample are missing.
-It is recommended to add a Table with the evaluation of the Methodological Quality.
-Table 2. Summary and characteristics of the studies included in the review (n = 5 studies) must be an error.
DISCUSSION
Strengths and Limitations
-LINES 267-268. Repeated information. This is not a strength, but a requirement of the type of study.
CONCLUSIONS
- The findings of this scoping review indicate positive responses in cardiovascular variables, which may help influence the decision-making of professionals….” Specify the type of professionals.
- “…prescribing exercise to cardiovascular responses in normotensive and hypertensive participants” specify type of exercise and type of hypertension. The results of 6 studies with such a small population sample cannot be extrapolated.
- “However, it is emphasized that longitudinal studies…” when talking about longitudinal studies, do they refer to a long-term follow-up or an observational study design? Specify.
Author Contributions: A publication of these characteristics does not require such a large number of authors.
Author Response
Reviewer 1
Major comments
Thank you for the opportunity to review this work that examines the cardiovascular responses after experimental conditions (Manual Massage or Foam Rolling) combined or not with Strength.
We appreciate.
KEYWORDS
It is recommended to use terms that appear in a thesaurus. This term “exercise cardiology” does not appear as such. It is recommended to replace “myofascial release” with the MESH term “Myofascial Release Therapy”.
We have revised this section, and it now reads as follows: “blood pressure; post-exercise hypotension; myofascial release therapy; manual therapy.”
INTRODUCTION
“These findings support a similar responsive hypothesis for the manual massage (MM) technique given the similarity in the application of both techniques (FR vs. MM – pressure and tissue sliding)”. This statement is not adequate. Massage techniques do not consist only of pressure and sliding on the tissue. They are more complex and varied maneuvers than those applied with Foam Rolling. Argue it in a better way.
We have revised this section, and it now reads as follows:
“The similarity of these findings suggests equivalent effects between FR and MM, despite the technical differences in their application. Among these differences, it is noteworthy that the superficial contact with the participant’s skin and the application of sliding pressure on the tissue appear to stimulate muscular and fascial mechanoreceptors, which exert inhibitory effects, ultimately leading to a reduction in muscle tone.”
On the other hand, it is recommended to provide more evidence on the mechanisms of manual massage on blood pressure.
We appreciate the suggestion; however, we believe that maintaining a stronger methodological framework requires presenting the mechanisms in our discussion to better support the interpretation of our findings (both in our study and in the existing literature).
MATERIALS AND METHODS
Stage 2: Identification of Relevant Studies: Was the search conducted on a single day, January 29, 2025? Who reproduced it, one or more people?
We appreciate you highlighting this point. The date indicated in the paper (January 29, 2025) corresponds to the completion of the article search within the selected databases. We’ve now revised the text to enhance clarity and address the request accordingly.
“Two authors (ERM and LMA) conducted searches in 4 databases (Nursing and Allied Health (CINAHL), Cochrane Library, PubMed®, and SciELO) between December 20th, 2024, and January 29th, 2025.”
Stage 3: Study Selection: justify this inclusion criterion “aged between 18 and 59 years”.
This age range was deliberately chosen to encompass all age groups within the adult population while excluding older adults. In Brazil, individuals are considered adults starting at the age of 18. The WHO defines "elderly" as people aged 60 or older, which is the same definition used in Brazil.
It is striking that only randomized controlled or counterbalanced crossover designs were included in the selection of studies. A scoping review should not be restrictive and should also include other study designs.
We have corrected this section for better understanding and it now reads: “Cross-Sectional studies (i.e., randomized controlled or cross-over trials) were included.”
Although scoping reviews cannot be registered in PROSPERO, it is recommended that the review protocol be freely available through platforms such as Figshare, Open Science Framework, ResearchGate, Research Square.
We appreciate your concern and suggestion. Our methodology, as well as the full text, is available in Pre-Print format.
RESULTS
Results on cardiac output, and arterial vascular perfusion are missing.
We’ve now made the necessary corrections to the results description to address this suggestion. The edits have been highlighted throughout the text for better visualization.
Make it clear whether 2 or 3 independent reviewers were involved in the selection, data extraction and methodological quality assessment phases.
We’ve now made the necessary corrections to the results description to address this suggestion. The edits have been highlighted throughout the text for better visualization.
Figure 1. Flow diagram. Improve the quality of the image because it is not displayed well. It does not correctly represent the article selection process. Duplicates are eliminated first. The reasons for exclusion based on Title or Abstract are not understood. There are numerical errors. Is the final sample 5 or 6 studies?.
We appreciate you highlighting this point. Articles were excluded based on their title if they did not mention the topics reviewed in this study (MM, FR, and ST on hemodynamic and autonomic responses). Additionally, articles were excluded based on the abstract if they described ineligible participants (according to our inclusion criteria), did not present comparator interventions, or reported ineligible outcomes. We have now made some edits to Figure 1 to enhance the visualization of the information.
Table 1. Characteristics of the participants (n = 6 studies). Describe better the characteristics of the studies that are part of the review. Data from the Sample are missing.
We appreciate this, but your request is not entirely clear. Each of our tables serves a specific purpose. Table 1 presents the main characteristics of the sample, while Table 2 provides methodological details and the results of each study included in the review. We have now revised the table contents based on the comments from all three reviewers.
It is recommended to add a Table with the evaluation of the Methodological Quality.
We have created a new table (Table 3) containing only the methodological quality information.
Table 2. Summary and characteristics of the studies included in the review (n = 6 studies) must be an error.
We have corrected this.
DISCUSSION
Strengths and Limitations
LINES 267-268. Repeated information. This is not a strength, but a requirement of the type of study.
We have corrected this.
CONCLUSIONS
The findings of this scoping review indicate positive responses in cardiovascular variables, which may help influence the decision-making of professionals….” Specify the type of professionals.
We have revised this section, and it now reads as follows:
“The results of this scoping review indicate positive responses in hemodynamic and autonomic variables, which may help inform decision-making for professionals prescribing exercise (e.g., Exercise Physiologists, Biomechanics, and Physical Therapists) to influence hemodynamic and autonomic responses in normotensive individuals.”
“…prescribing exercise to cardiovascular responses in normotensive and hypertensive participants” specify type of exercise and type of hypertension. The results of 6 studies with such a small population sample cannot be extrapolated.
We have revised this section, and it now reads as follows:
“The results of this scoping review indicate positive responses in hemodynamic and autonomic variables, which may help inform decision-making for professionals prescribing exercise (e.g., Exercise Physiologists, Biomechanics, and Physical Therapists) to influence hemodynamic and autonomic responses in normotensive individuals.”
“However, it is emphasized that longitudinal studies…” when talking about longitudinal studies, do they refer to a long-term follow-up or an observational study design? Specify.
We have revised this section, and it now reads as follows:
“Although the investigations included in this review were acute, the observed data suggest that both MM and FR can be powerful tools for improving hemodynamic and autonomic aspects. However, it is emphasized that long-term and follow-up studies are essential for a better understanding of these parameters.”
Author Contributions: A publication of these characteristics does not require such a large number of authors.
We appreciate the concern on this point; however, we emphasize that all authors of this review have contributed significantly to its development and fully support their participation.
Reviewer 2 Report
Comments and Suggestions for Authors
Dear Editor, I would like to thank you for your kind invitation to review the article entitled “Effect of Manual Massage, Foam Rolling, and Strength Training on Cardiovascular Responses in Adults: A Scoping Review”.
The purpose of this investigation is to review the cardiovascular responses after experimental conditions of massage – manual (MM) or with foam rolling (FR) – combined or not with strength training in healthy adults
I think the subject of the study is quite interesting. I have some suggestions for the authors to improve the article.
Introduction
In this section, it should be emphasized more clearly for the readers why these methods are discussed and their effects are to be examined.
Results
In the picture containing the flow chart, it is seen that a total of 5 articles are included. However, it is stated in the table and content that 6 articles are included in the study. This error needs to be corrected.
Table – 2 can be simplified or shortened for easier reading. Perhaps, if the Pedro score results are removed from this table, the table can be shortened. However, this preference is up to the authors.
The results can also be presented to the readers in plain text.
Discussion
The high quality of the included studies can be considered a strength of your study.
It may be useful for readers to present the results of the studies a little more in the discussion section.
I also think it is important to include in the discussion section how the results of this meta-analysis study will benefit people working in clinics with your own perspective and experiences.
I congratulate the authors for this study and wish them success in their work.
Author Response
Major comments
Dear Editor, I would like to thank you for your kind invitation to review the article entitled “Effect of Manual Massage, Foam Rolling, and Strength Training on Cardiovascular Responses in Adults: A Scoping Review”.
The purpose of this investigation is to review the cardiovascular responses after experimental conditions of massage – manual (MM) or with foam rolling (FR) – combined or not with strength training in healthy adults.
I think the subject of the study is quite interesting. I have some suggestions for the authors to improve the article.
INTRODUCTION
In this section, it should be emphasized more clearly for the readers why these methods are discussed and their effects are to be examined.
We have revised this section, and it now reads as follows:
“Understanding the effects of these techniques (FR and MM), whether applied in isolation or combined with RT, helps bridge gaps in exercise prescription and enhances the plausibility of their integration into treatment routines (through movement) targeting autonomic and hemodynamic responses. However, there is limited evidence in the literature regarding the blood pressure response to different combinations of FR and ST.”
RESULTS
In the picture containing the flow chart, it is seen that a total of 5 articles are included. However, it is stated in the table and content that 6 articles are included in the study. This error needs to be corrected.
We have revised this section, thank you.
Table – 2 can be simplified or shortened for easier reading. Perhaps, if the Pedro score results are removed from this table, the table can be shortened. However, this preference is up to the authors.
Table 2 was revised and the PEDro classification changed to Table 3.
The results can also be presented to the readers in plain text.
We’ve now made the necessary corrections to the results description to address this suggestion. The edits have been highlighted throughout the text for better visualization.
DISCUSSION
The high quality of the included studies can be considered a strength of your study.
Thank you. We have revised this section, and it now reads as follows:
“Finally, all articles included in the results of this review demonstrate moderate to high methodological quality (Table 3).”
It may be useful for readers to present the results of the studies a little more in the discussion section.
We’ve now made the necessary corrections to the results description to address this suggestion. The edits have been highlighted throughout the text for better visualization.
I also think it is important to include in the discussion section how the results of this meta-analysis study will benefit people working in clinics with your own perspective and experiences.
We’ve now made the necessary corrections to the conclusion description to address this suggestion. The edits have been highlighted throughout the text for better visualization.
I congratulate the authors for this study and wish them success in their work.
Thank you.
Reviewer 3 Report
Comments and Suggestions for Authors
The article presents a well-founded theoretical basis and a logical flow of ideas. However, some sections are excessively long and could be reorganized to improve readability. The abstract could be more concise and direct, highlighting the main quantitative findings rather than only describing the general effects of the interventions. The introduction is well-supported, but the justification for the review could be clearer. What specific gap in the literature does this study aim to address?
Methodology
The study selection criteria are well described, but it is unclear whether any quality assessment beyond the PEDro scale was conducted. Adding a risk of bias evaluation would enhance the reliability of the conclusions.
Only six studies were selected, raising concerns about the robustness of the conclusions.
Analysis of Results
The presentation of findings could be better structured, organizing the evidence more comparatively across the included studies. A comparative table summarizing the study design, sample characteristics, and key findings would be beneficial.
The results on blood pressure and hemodynamics are promising. Still, the discussion lacks sufficient detail on the potential physiological mechanisms explaining the differences between interventions (e.g., neurophysiological effects of foam rolling vs. manual massage).
Some inferences are made without a clear explanation of the causality between variables. For example, the reduction in blood pressure following massage and foam rolling could be influenced by multiple factors, and the article could better discuss the variability of these effects across studies.
Discussion
The discussion effectively links the findings to existing literature but could include a dedicated section on study limitations and suggestions for future research.
Some arguments rely on indirect references (systematic reviews rather than primary studies). Incorporating more direct references would strengthen the argumentation.
There is limited discussion on the duration of the observed effects. It would be important to consider whether the benefits are transient or sustainable over time.
Conclusion
The conclusion could be more concise, reinforcing the key findings and their practical implications for exercise and manual therapy prescriptions.
While the article mentions the need for longitudinal studies, it does not specify which aspects of the relationship between massage, foam rolling, and resistance training require further investigation. Providing concrete suggestions for future research would strengthen the closing section.
Reformatting lengthy sections, particularly in the introduction and discussion, to enhance readability and conciseness.
Author Response
Major comments
The article presents a well-founded theoretical basis and a logical flow of ideas. However, some sections are excessively long and could be reorganized to improve readability. The abstract could be more concise and direct, highlighting the main quantitative findings rather than only describing the general effects of the interventions. The introduction is well-supported, but the justification for the review could be clearer. What specific gap in the literature does this study aim to address?
Thank you. We’ve now made the necessary corrections to the introduction description to address this suggestion. The edits have been highlighted throughout the text for better visualization.
METHODOLOGY
The study selection criteria are well described, but it is unclear whether any quality assessment beyond the PEDro scale was conducted. Adding a risk of bias evaluation would enhance the reliability of the conclusions.
We have now revised this section and included Table 3, which provides a detailed breakdown of the PEDro assessment criteria. We appreciate the suggestion; however, we have chosen not to include the risk of bias assessment, as this is a scoping review.
Only six studies were selected, raising concerns about the robustness of the conclusions.
We have now adjusted the tone of the conclusion to make it more neutral. However, we emphasize that, despite being considered a small number, the six articles included represent the entirety of studies available on this topic.
ANALYSIS OF RESULTS
The presentation of findings could be better structured, organizing the evidence more comparatively across the included studies. A comparative table summarizing the study design, sample characteristics, and key findings would be beneficial.
Thank you for the suggestion. We have now reorganized the presentation of the results and included a new table (Table 3) detailing the characteristics of the PEDro assessment point by point.
The results on blood pressure and hemodynamics are promising. Still, the discussion lacks sufficient detail on the potential physiological mechanisms explaining the differences between interventions (e.g., neurophysiological effects of foam rolling vs. manual massage).
The current scientific literature, which focuses on discussing potential physiological mechanisms, tends to consistently indicate similar effects, regardless of whether the technique is passive or active. This reinforces the possibility that the responses are linked to therapeutic touch, a concept that has been incorporated throughout the text. Introducing additional mechanisms beyond those discussed in the discussion section does not seem to enhance the strength of the text but rather makes it more speculative, given that this review does not aim to explore such points (neurophysiological mechanisms).
Some inferences are made without a clear explanation of the causality between variables. For example, the reduction in blood pressure following massage and foam rolling could be influenced by multiple factors, and the article could better discuss the variability of these effects across studies.
We’ve now made the necessary corrections to the text to address this suggestion. All edits have been highlighted throughout the text for better visualization.
DISCUSSION
The discussion effectively links the findings to existing literature but could include a dedicated section on study limitations and suggestions for future research.
We’ve now made the necessary corrections to the text to address this suggestion. All edits have been highlighted throughout the text for better visualization.
Some arguments rely on indirect references (systematic reviews rather than primary studies). Incorporating more direct references would strengthen the argumentation.
We’ve now made the necessary corrections to the text to address this suggestion. All edits have been highlighted throughout the text for better visualization.
There is limited discussion on the duration of the observed effects. It would be important to consider whether the benefits are transient or sustainable over time.
We’ve now made the necessary corrections to the text to address this suggestion. All edits have been highlighted throughout the text for better visualization.
CONCLUSION
The conclusion could be more concise, reinforcing the key findings and their practical implications for exercise and manual therapy prescriptions.
We’ve now made the necessary corrections to the text to address this suggestion. All edits have been highlighted throughout the text for better visualization.
While the article mentions the need for longitudinal studies, it does not specify which aspects of the relationship between massage, foam rolling, and resistance training require further investigation. Providing concrete suggestions for future research would strengthen the closing section.
We’ve now made the necessary corrections to the text to address this suggestion. All edits have been highlighted throughout the text for better visualization.
Reformatting lengthy sections, particularly in the introduction and discussion, to enhance readability and conciseness.
Thank you. We’ve now made the necessary corrections to the text to address this suggestion. All edits have been highlighted throughout the text for better visualization.
Round 2
Reviewer 1 Report
Comments and Suggestions for Authors
-Stage 3: Study Selection: justify this inclusion criterion “aged between 18 and 59 years”.
This age range was deliberately chosen to encompass all age groups within the adult population while excluding older adults. In Brazil, individuals are considered adults starting at the age of 18. The WHO defines "elderly" as people aged 60 or older, which is the same definition used in Brazil.
Include justification as a criterion.
-It is striking that only randomized controlled or counterbalanced crossover designs were included in the selection of studies. A scoping review should not be restrictive and should also include other study designs.
We have corrected this section for better understanding and it now reads: “Cross-Sectional studies (i.e., randomized controlled or cross-over trials) were included.”
I appreciate your work on the review, but I insist that scoping reviews should expand study designs for inclusion. If this is not possible, they should indicate this limitation in the Discussion.
-Although scoping reviews cannot be registered in PROSPERO, it is recommended that the review protocol be freely available through platforms such as Figshare, Open Science Framework, ResearchGate, Research Square.
We appreciate your concern and suggestion. Our methodology, as well as the full text, is available in Pre-Print format.
Registering a systematic review on these platforms ensures transparency and avoids duplication of effort, serving as a prerequisite for publication. A preprint, however, involves sharing the full text before formal publication. Both are useful, but serve different purposes.
-Figure 1. Flow diagram. Improve the quality of the image because it is not displayed well. It does not correctly represent the article selection process. Duplicates are eliminated first. The reasons for exclusion based on Title or Abstract are not understood. There are numerical errors. Is the final sample 5 or 6 studies?.
We appreciate you highlighting this point. Articles were excluded based on their title if they did not mention the topics reviewed in this study (MM, FR, and ST on hemodynamic and autonomic responses). Additionally, articles were excluded based on the abstract if they described ineligible participants (according to our inclusion criteria), did not present comparator interventions, or reported ineligible outcomes. We have now made some edits to Figure 1 to enhance the visualization of the information.
The eligibility process is still unclear. Why are there 6 articles in the end?